# Uranium transport in acidic brines under reducing conditions

Alexander Timofeev[1], Artaches A. Migdisov[2], Anthony E. Williams-Jones[1], Robert Roback[2], Andrew T. Nelson[3] & Hongwu Xu[2]

The behavior of uranium in environments, ranging from those of natural systems responsible for the formation of uranium deposits to those of nuclear reactors providing 11% of the world's electricity, is governed by processes involving high-temperature aqueous solutions. It has been well documented that uranium is mobile in aqueous solutions in its oxidized, $U^{6+}$ state, whereas in its reduced, $U^{4+}$ state, uranium has been assumed to be immobile. Here, we present experimental evidence from high temperature (>100 °C) acidic brines that invalidates this assumption. Our experiments have identified a new uranium chloride species $(UCl_4°)$ that is more stable under reducing than oxidized conditions. These results indicate that uranium is mobile under reducing conditions and necessitate a re-evaluation of the mobility of uranium, particularly in ore deposit models involving this metal. Regardless of the scenario considered, reducing conditions can no longer be considered a guarantee of uranium immobility.

[1] Department of Earth & Planetary Sciences, McGill University, 3450 University Street, Montreal, QC H3A 0E8, Canada. [2] Earth and Environmental Division, Los Alamos National Laboratory, P.O. Box 1663, M.S. J535, Los Alamos, NM 87545, USA. [3] Materials Science and Technology Division, Los Alamos National Laboratory, P.O. Box 1663, M.S. E549, Los Alamos, NM 87545, USA. Correspondence and requests for materials should be addressed to A.T. (email: alexander.timofeev@mail.mcgill.ca)

From source to sink, the nuclear applications and associated chemistry of uranium have been extensively investigated. As most processes of enrichment and uses of uranium involve its contact, either direct or indirect, with aqueous solutions, many studies have focused on its interactions with high temperature (100 to ≥400 °C) aqueous fluids. However, our understanding of the behavior of uranium has been governed by the belief that uranium in its $U^{4+}$ state cannot be present in significant concentrations (>ppt) in aqueous fluids[1]. In other words, uranium is immobile under reducing conditions. Thus, hydrothermal models for the formation of uranium deposits call for the mobilization of the metal as $U^{6+}$ and its precipitation as a $U^{4+}$ mineral, e.g., uraninite, due to a reduction in oxygen fugacity ($fO_2$)[2,3,].

The accepted model for the formation of unconformity-type uranium deposits involves the interaction of an oxidized basinal brine, which transports the uranium as $U^{6+}$, with a reducing, graphite-bearing metapelite that leads to the precipitation of uraninite or the $U^{4+}$-bearing phase, pitchblende[2,3]. However, evidence collected from magnetite group iron oxide copper–gold (IOCG) deposits, which are another important source of uranium, suggests that, whereas this paradigm satisfies observations made at low-to-moderate temperatures, e.g., for unconformity-type uranium deposits, it likely fails for temperatures greater than 250 °C[4]. In order to explain the formation of the high temperature IOCG-type deposits, it is necessary to postulate uranium transport under reducing conditions.

In nuclear waste management, there is an underlying assumption that reducing conditions will prevent uranium migration[5–7]. This extends also to catastrophic accidents, such as that at the Fukushima power plant, for which it is believed that reducing conditions will inhibit uranium migration. A lack of thermodynamic data for uranium species at elevated temperatures is a key factor preventing accurate predictions of uranium behavior during such an event[8,9]. These and other conclusions about the behavior of uranium in the presence of aqueous fluids, whether it be for nuclear industry applications or models of uranium ore formation, routinely draw upon the dogma that "uranium is immobile in the reduced state".

Despite the importance of understanding the behavior of uranium in high temperature (>100 °C) fluids, there have been very few studies of uranium mineral solubility at elevated temperature, particularly under reducing conditions. Laboratory and field-based observations indicate that oxidizing, oxygen rich, fluids are capable of transporting considerable (ppm) concentrations of this metal as an aqueous species[10–12]. By contrast, studies of uranium behavior in reduced fluids have either not defined the exact oxygen fugacity of the fluid (i.e., the accurate redox state of the solution) or neglected to consider the impact of common ligands in the fluid such as chloride, fluoride, sulfate, and carbonate[13–15]. Such ligands may enhance the solubility of metals due to the formation of metal complexes (e.g., $UO_2CO_3°$)[16]. At high temperatures their impact on uranium mobility is uncertain. However, both oxygen fugacity and ligand concentration must be considered in order to accurately evaluate uranium mobility.

The most abundant ligand in seawater[17], most ore-forming fluids[18], and many waters surrounding underground waste repositories[19] is chloride ($Cl^-$), which can attain weight percent concentrations in these fluids. Because of the ubiquitous presence of chloride in natural systems, results of measurements of the solubility of $UO_3·nH_2O$ in $NaCl–H_2O$ solutions[12] and spectroscopic data suggesting that U–Cl complex formation may be important at elevated temperature[20,21], we have made U–Cl speciation the focus of this paper. We also do so because of observations of uranium- and chloride-rich fluid inclusions, suggesting that the impact of chloride may be underestimated in current models of uranium ore formation[12]. In this study, we identify a high temperature U–Cl species that explains these observations and extract thermodynamic data for this species from experiments conducted at variable chloride and $fO_2$ conditions. As the thermodynamic properties of uranium minerals such as uraninite are known, we use this information in conjunction with the data for the U–Cl species to predict uranium behavior in hydrothermal fluids for a range of reducing conditions in which uranium mobility has previously been discounted.

## Results

**The U–Cl species**. To determine the stability of U–Cl complexes, we conducted solubility experiments in titanium autoclaves (Supplementary Figures 1 and 2, and Supplementary Table 1). Acidic solutions (pH 25 °C ~2) having variable NaCl concentrations (0.1−1m) were equilibrated with a synthetic uranium oxide powder (Supplementary Figure 3) at temperatures ranging from 250 to 350 °C and saturated water vapor pressure. A $MoO_2–MoO_3$, and a Ni–NiO or Co–CoO buffer, all having no contact with the experimental solution were employed to maintain either oxidizing or reducing conditions, respectively (Supplementary Table 2). Changes in the dissolved uranium concentrations with increases in chloride activity were used to identify corresponding U–Cl species. Initial experiments with the Ni–NiO buffer (reducing conditions) employed the uranium oxide $UO_2^{cryst}$. These experiments yielded a dependence of the log activity of uranium on the log activity of chloride with a slope of four (Fig. 1a), which is consistent with the dissolution reaction:

$$UO_2^{cryst} + 4Cl^- + 4H^+ = UCl_4° + 2H_2O \qquad (1)$$

The dissolved uranium species at reducing conditions was not known a priori to contain either uranium in the reduced, $U^{4+}$ form, or in its oxidizing, $U^{6+}$ state, although we assume the former in the $fO_2$ independent Reaction (1). If the uranium was in the 6+ state, the uranium dissolution reaction would be $fO_2$ dependent. To determine if this was the case, we conducted additional experiments with a Co–CoO buffer and a $UO_2^{cryst}$ solid. The results of these experiments were well within the experimental uncertainty of the data obtained using the Ni–NiO buffer, thereby supporting our interpretation of the species at reducing conditions as being $UCl_4°$, and validating Reaction (1) (Fig. 1a). The rapid increase in stability with increasing temperature of this newly discovered species leads to concentrations of uranium that reach ppm levels (e.g., ~4 ppm at 400 °C with a $Cl^- = 0.10$ and a pH of 2.5) (Fig. 1a).

In addition, we conducted experiments under reducing conditions with $U_3O_8^{cryst}$ so as to assess the impact of $U^{6+}$ in the solid, for example as $U_3O_8^{cryst}$ inclusions in pitchblende, on uranium mobility in a reducing environment (Fig. 1b). The reaction inferred from these results is similar to Reaction (1):

$$U_3O_8^{cryst} + 12Cl^- + 12H^+ = 3UCl_4° + 6H_2O + O_2 \qquad (2)$$

The data from these experiments also allowed us to compare the thermodynamic data and formation constants obtained for $UCl_4°$ from Reaction (1) with those obtained from Reaction (2). The two sets of data were found to be in very good agreement (Table 2). Under oxidizing conditions, the logarithm of uranium activity has a slope of two with respect to log chloride activity at both 300 and 350 °C (Fig. 1b). The dissolution reaction for the species that best fits this trend is:

$$2U_3O_8^{cryst} + 12Cl^- + 12H^+ + O_2 = 6UO_2Cl_2° + 6H_2O \qquad (3)$$

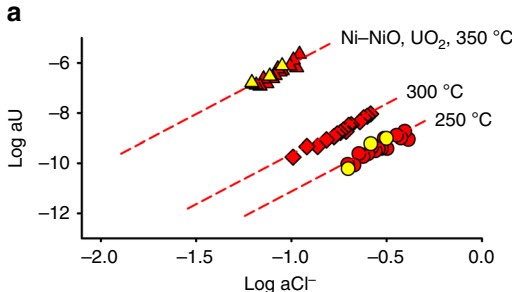

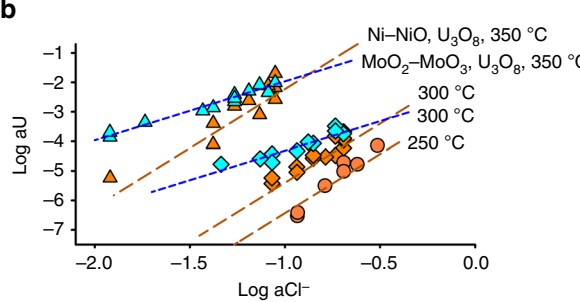

**Fig. 1** The activity of uranium determined experimentally to be in equilibrium with uranium oxide solids. **a** Uranium activity as a function of chloride activity for experiments conducted using $UO_2^{cryst}$ under reducing conditions. Red circles, diamonds, and triangles indicate results obtained at 250, 300, and 350 °C, respectively. The slope of all trend lines is 4. The yellow circle and triangle symbols indicate use of the Co-CoO buffer at 300 and 350 °C. The results of these experiments are indistinguishable from those utilizing the Ni-NiO buffer. All data are normalized to a pH of 2.5 using the stoichiometry of Reactions (1), (2), and (3). The trend lines represent the best fits to the data, and were calculated using the equilibrium constants for the speciation reaction (Table 1). **b** Uranium activity determined from solubility experiments employing $U_3O_8^{cryst}$ plotted for specified temperature (circles—250 °C, diamonds—300 °C, triangles—350 °C) and oxygen fugacity (oxidizing—blue; reducing—brown) as a function of chloride activity. The slope of the fit under oxidizing conditions is 2, and under reducing conditions it is 4. Trend lines for data representing oxidizing conditions were calculated using equilibrium constants of 39.6 and 50.9 at 300 and 350 °C, respectively. They were fitted independently of those described in ref. [21].

On the basis of these results, we conclude that uranium speciation in chloride-bearing solutions at elevated temperature is dominated by $UO_2Cl_2°$ and $UCl_4°$ under oxidizing and reducing conditions, respectively.

**Formation constants**. In order to make our data accessible for thermodynamic modeling, we calculated formation constants (logß), which are independent of the experimental uranium oxide solid, for $UO_2Cl_2°$ and $UCl_4°$. The formation constant reaction for $UO_2Cl_2°$ is written as follows:

$$UO_2^{2+} + 2Cl^- = UO_2Cl_2°$$
$$logß_1 = \log a_{UO_2Cl_2^0} - \log a_{UO_2^{2+}} - 2\log a_{Cl^-} \quad (4)$$

We calculated the formation constants for Reaction (4) to be $4.29 \pm 0.08$ at 300 °C and $7.21 \pm 0.07$ at 350 °C. These values are in good agreement with those obtained independently using ultraviolet-visible spectroscopy (UV-VIS) measurements at temperatures ≤250 °C (Fig. 2)[21]. The corresponding reaction for

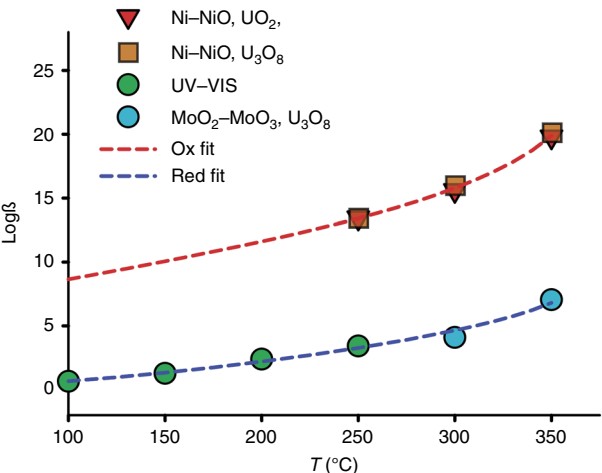

**Fig. 2** Comparison of formation constants determined in this study to those previously reported. Formation constants for $UCl_4°$ at reducing conditions (Ni-NiO oxygen fugacity buffer) were fitted to the Bryzgalin–Ryzhenko model (red dashed line). Regardless of whether the experiments used $UO_2^{cryst}$ or $U_3O_8^{cryst}$, the resulting formation constants are indistinguishable. Uncertainties for the formation constants are smaller than the symbols. A similar fit for $UO_2Cl_2°$ at oxidizing conditions (blue dashed line) demonstrates that the data obtained in this study (blue circles) are consistent with those of ref. [21] (green circles)

$UCl_4°$ is:

$$U^{4+} + 4Cl^- = UCl_4°　　logß_2 = \log a_{UCl_4^0} - \log a_{U^{4+}} - 4\log a_{Cl^-}$$
$$(5)$$

We had two data sets to choose from for our calculation of logß_2. The formation constants calculated using data from Reaction (1) (logß_1) are $13.4 \pm 0.05$, $15.6 \pm 0.06$, and $19.8 \pm 0.06$ for 250, 300, and 350 °C respectively. The values of logß_2 calculated using $U_3O_8^{cryst}$ are $13.5 \pm 0.08$ at 250 °C, $16.0 \pm 0.08$ at 300 °C, and $20.0 \pm 0.14$ at 350 °C (Fig. 2), which are very close to those obtained using Reaction (1) (see above). This confirms the dominance of $UCl_4°$ in the solutions. The presence of a neutral species is consistent with the observation that, with increasing temperature, the dipole moment of water decreases and its hydrogen bonding network weakens considerably, thereby favoring aqueous species of low or no charge[22].

## Discussion

It is clear from the results of our experiments that considerable uranium transport can take place under reducing conditions. If the chemical composition of a reducing, chloride-bearing fluid is fixed, an increase in temperature from 250 to 350 °C will result in a five orders of magnitude rise in uranium solubility. Raising the chloride activity of the fluid by one logarithmic unit or decreasing the pH by one unit will increase uranium solubility by four orders of magnitude (Fig. 1a). Comparable, but less pronounced effects are evident for oxidizing conditions; the aforementioned changes in chloride activity and pH result in increases of uranium solubility by two rather than four orders of magnitude (Fig. 1b). In short, the ideal fluid for moving uranium is hot, chloride-rich, and highly acidic. The opposite is true for uranium removal from a fluid. This could be accomplished by cooling (fluids at ≤150 °C are $U^{4+}$ poor[23]), a reduction in chloride concentration or increasing the pH of the fluid. Any one or more of these changes are common to both man-made and natural uranium-bearing systems. An important example of the latter is provided by the

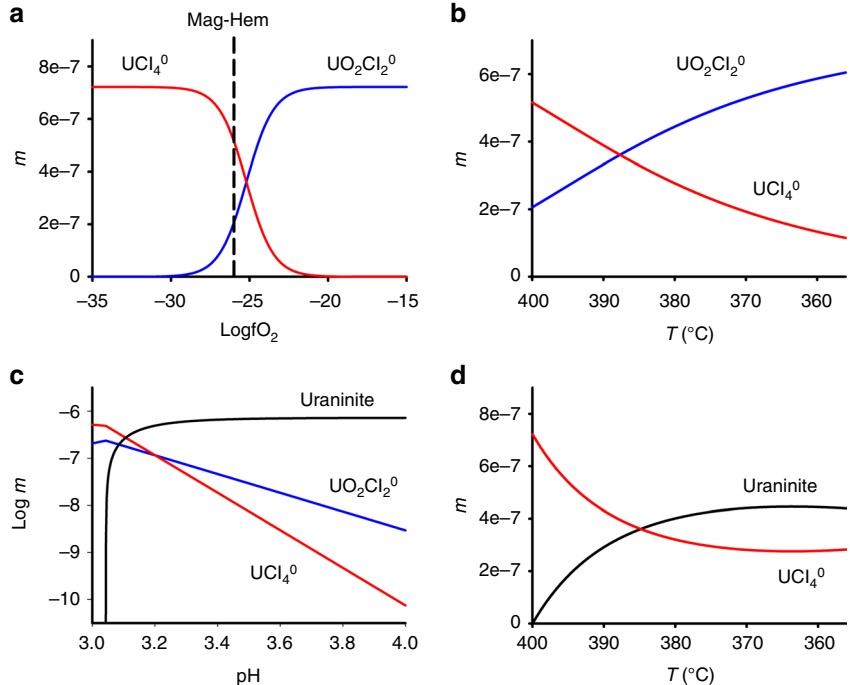

**Fig. 3** A model of uranium behavior in a magnetite-bearing IOCG system. **a**, **b** Raising the oxygen fugacity of a 2*m* NaCl acidic brine at 400 °C and 500 bar does not induce uraninite precipitation (**a**), nor does cooling under oxidizing conditions (**b**). **c**, **d** However, increasing the pH at oxidizing conditions (**c**) or decreasing the temperature of a reduced solution (**d**) triggers uraninite deposition

hot, saline brine responsible for the formation of magnetite- and uranium-bearing iron oxide copper–gold (IOCG) deposits within which, at depth, a predominance of magnetite in observed alteration assemblages demonstrates reducing conditions[4]. We chose this setting to illustrate the importance of these newly identified U–Cl species.

We constructed a model to evaluate uranium transport by a magmatic fluid under reducing conditions in an IOCG setting. With formation constants for the aqueous species in hand, we fitted our reducing (Fig. 2) and oxidizing data to the Bryzgalin–Ryzhenko equation of state[24]. We then saturated the fluid (a 2*m* NaCl acidic brine) with respect to uraninite at 400 °C, 500 bar and reducing conditions (log$fO_2$ = −35, ~QFM-4) and determined that this fluid is capable of dissolving ~0.27 ppm of the $UCl_4^0$ species. This sizable amount of uranium must be precipitated as uraninite ($UO_2^{cryst}$) in order to form an ore deposit. Mixing of the reduced fluid with an oxidizing (log$fO_2$ = −26), but otherwise identical fluid does not lead to uraninite precipitation, but rather a change in speciation (Fig. 3a). Indeed, at the magnetite–hematite $fO_2$ buffer (log$fO_2$ = −26), corresponding to shallower, more oxidizing parts of IOCG systems, $UCl_4°$ remains the predominant aqueous species in solution. Cooling of an oxidizing fluid results in a similar and gradual change in speciation, but no uraninite precipitation (Fig. 3b). By contrast, increasing the pH of the oxidizing solution through interaction with the wall rocks or mixing with meteoric waters rapidly removes the dissolved uranium (Fig. 3c). At reducing (log$fO_2$ = −35) conditions, a decrease in temperature promotes uraninite deposition (Fig. 3d). In summary, the model outlined above clearly identifies mechanisms by which uranium transport and deposition can both occur under reducing conditions in natural systems.

Reducing conditions in uranium-bearing systems are not limited to IOCG deposits and some models explaining the formation of unconformity-type uranium deposits involve reduced basement rocks containing uranium oxides[25]. Moreover, our data

suggest that seawater used to cool fuel assemblies after the Fukushima nuclear accident, which likely remained at temperatures near or above boiling for months following the event[8,9] would have the potential to dissolve and mobilize uranium at ppm concentrations. These and other natural and anthropogenic scenarios involving uranium transport under reducing conditions, for which our study represents a starting point, require further investigation. It is also important to note that ligands other than chloride may play an important role in reduced uranium transport. Hard-soft-acid-base principles suggest that uranium, a hard (high charge to radius ratio) cation, should form strong complexes with fluoride and carbonate ions, which are both hard anions[26]. Indeed, both ions have been proposed as ligands likely to form complexes with uranium in IOCG systems[27]. The presence of these additional ligands in hydrothermal brines will further promote the dissolution of uranium under reducing conditions. This study represents a first attempt to identify and determine thermodynamic constants for U–Cl species under reducing conditions at elevated temperature, and to evaluate the potential for uranium mobility under such conditions. The new species identified in this study may be just the first of many.

## Methods
**Experimental setup and procedure**. The experiments involved measuring the solubility of synthetic $U_3O_8$ and $UO_2$ (solid) in aqueous solutions of variable NaCl concentration at temperatures of 250, 300, and 350 °C, and vapor-saturated water pressure. All the experiments with $U_3O_8$ (solid), and those with $UO_2$ (solid) at 300 and 350 °C, were performed in titanium autoclaves (Supplementary Figure 1). Experiments with $UO_2$ (solid) at 250 °C were conducted in titanium autoclaves containing a Teflon liner that limited contact of the experimental solution with the autoclave body (Supplementary Figure 2). The consistency of the results of the Teflon-lined experiments with those of the Teflon-free runs demonstrated that the extent of reaction between the titanium autoclaves and the experimental solution was negligible and did not affect the outcome of the experiments (Fig. 2 and Supplementary Table 1). Experiments with Teflon could not be conducted at 300 and 350 °C as perfluoroalkoxy (PFA) Teflon has a maximum operating temperature of ~260 °C[28].

**Table 1 Logarithms of equilibrium constants and their associated 1σ uncertainty for uranium oxide dissolution reactions**

| T (°C) | 250 | 300 | 350 |
|---|---|---|---|
| Oxidizing ($MoO_2$–$MoO_3$ buffer) | | | |
| $2U_3O_8{}^{cryst} + 12Cl^- + 12H^+ + O_2 = 6UO_2Cl_2° + 6H_2O$ | — | 42.3 ± 0.96 | 49.6 ± 0.72 |
| Reducing (Ni–NiO buffer) | | | |
| $U_3O_8{}^{cryst} + 12Cl^- + 12H^+ = 3UCl_4° + 6H_2O + O_2$ | −18.0 ± 0.72 | −10.5 ± 0.60 | 3.01 ± 1.02 |
| $UO_2{}^{cryst} + 4Cl^- + 4H^+ = UCl_4° + 2H_2O$ | 2.88 ± 0.17 | 4.37 ± 0.07 | 7.96 ± 0.14 |

The equilibrium constants are grouped according to whether they were from oxidizing ($MoO_2$–$MoO_3$) or reducing (Ni–NiO) oxygen fugacity buffer experiments. Data from ref.[21] for 100−250 °C were included during the optimization of $UO_2Cl_2°$. The dash indicates that data were not collected for this temperature.

**Table 2 Calculated formation constants (logß) for the uranium species identified in this study**

| T (°C) | 250 | 300 | 350 |
|---|---|---|---|
| $UO_2{}^{2+} + 2Cl^- = UO_2Cl_2°$ ($U_3O_8{}^{cryst}$) | — | 4.29 ± 0.08 | 7.21 ± 0.07 |
| $U^{4+} + 4Cl^- = UCl_4°$ ($U_3O_8{}^{cryst}$) | 13.5 ± 0.08 | 16.0 ± 0.08 | 20.0 ± 0.14 |
| $U^{4+} + 4Cl^- = UCl_4°$ ($UO_2{}^{cryst}$) | 13.4 ± 0.05 | 15.6 ± 0.06 | 19.8 ± 0.06 |

The formation constants are separated according to experiments conducted with different uranium oxide solids, which are indicated in brackets. The dash indicates that experiments were not conducted for this temperature

Chloride concentrations were controlled by dissolving known amounts of NaCl in nano-pure water. The pH of each solution was then reduced to the level of interest by adding an appropriate amount of trace metal grade HCl. The pH (25 °C) of the solutions used for the $U_3O_8{}^{cryst}$ experiments ranged from 2.2 to 2.6 prior to the start of the experiments. Solutions containing higher chloride concentrations experienced a more rapid increase in pH with increasing temperature than those of lower salinity. Therefore, high salinity solutions were created with a lower pH so as to have similar acidity at the temperatures of interest. The pH (25 °C) for the experiments with $UO_2{}^{cryst}$ was kept constant at ~1.5.

Experiments were initiated by placing small quartz holders containing uranium oxide powder in the titanium autoclaves, to which 15−18 ml of experimental solution was added. Experimental solution volumes were calculated to prevent solution contact with the inside of the oxygen fugacity buffer holder during volume increases associated with heating (~20% at 350 °C and saturated water vapor pressure). The quartz holders contained fine quartz wool above the uranium oxide solid to prevent any back-reaction during cooling of the experiment. A quartz holder just shorter than the height of the autoclave containing the oxygen fugacity buffers was then added to the autoclave. Oxidizing experiments employed $MoO_2$ and $MoO_3$ powders placed one atop the other and covered by fine quartz wool (Supplementary Figure 1), whereas reducing experiments used three strands of nickel metal wire (Supplementary Figure 2). The nickel wires were replaced for every experiment and during the course of the experimental run reacted with oxygen to form nickel oxide, thereby producing the Ni–NiO buffer. A small number of additional experiments with $UO_2{}^{cryst}$ were conducted with a reducing Co–CoO buffer at 250 °C. The log $fO_2$ values for the buffer equilibria at 250 °C–350 °C are shown in Supplementary Table 2. Prior to closure of the autoclave, high purity argon was flushed through the experimental solution for 30 min to significantly reduce the oxygen contained therein. After sealing the autoclave with a flexible carbon ring, the autoclave was heated in a Fisher Isotemp forced draft oven to the temperature of interest. An aluminum box lined the oven to reduce temperature gradients. The duration of each experiment was 7 days, which allowed the experiments to reach equilibrium.

Following completion of an experiment, the autoclave was air-cooled with a fan for 20 min and then unsealed. After removing the buffer and uranium oxide holders, an aliquot of the experimental solution was removed for measurement of the solution pH. For experiments with $UO_2{}^{cryst}$, the pH of the experimental solution was measured directly in the autoclave. To dissolve any potential precipitates formed during cooling, 5 ml of trace metal grade nitric acid was added to the autoclave and allowed to remain there for a minimum of 30 min.

The pH of the aliquots and the solutions of $UO_2{}^{cryst}$-based experiments was measured potentiometrically using a calibrated glass electrode. The concentrations of dissolved uranium were analyzed by inductively coupled plasma mass spectrometry (ICP-MS). The activity of chloride and the pH of each experimental solution was then recalculated at the temperature of the experiment using the

initial composition of the solution and the measured pH, respectively. Following completion of the experiments, samples of uranium oxide powder from oxidizing ($MoO_2$–$MoO_3$) and reducing (Ni–NiO) experiments were analyzed using X-ray diffraction (XRD) to verify the continued presence of $U_3O_8{}^{cryst}$ in experiments using this uranium oxide solid. The resulting XRD spectra confirmed the presence of $U_3O_8{}^{cryst}$ (Supplementary Figure 3).

**Data optimization**. The uranium species responsible for the observed uranium concentrations were recognized on the basis of trends observed in the data, specifically the slope of the logarithm of uranium activity with respect to the logarithm of chloride activity. As stated in the main text, the two species identified in this study are $UO_2Cl_2°$ and $UCl_4°$.

The excel optimization code entitled "OptimA" included as part of the Hch software package of ref.[29] was used to calculate standard Gibbs free energies of the two uranium species. Inputs to the code included the NaCl and HCl molality of every experiment, in addition to the molality of uranium analyzed in each experiment using ICP-MS. The molality of HCl of each experiment was calculated on the basis of the starting NaCl concentration and the pH measured after each experiment. Excess amounts of the relevant uranium oxide solid ($UO_2{}^{cryst}$ or $U_3O_8{}^{cryst}$) were included in the optimization, as well as the relevant oxygen fugacity buffers ($MoO_2$, $MoO_3$, or Ni, NiO). Aqueous species considered during the optimization were $H^+$, $OH^-$, $Na^+$, NaOH, NaCl, $Cl^-$, and HCl. The activity of the ions was calculated using the extended Debye–Hückel equation[30]:

$$\log \gamma_n = -\frac{A \cdot [z_n]^2 \cdot \sqrt{I}}{1 + B \cdot \r{a} \cdot \sqrt{I}} + b_\gamma \cdot I \qquad (6)$$

with A and B being the parameters of the Debye–Hückel equation, $b_\gamma$ the extended parameter, which depends on the nature of the background electrolyte, å the distance of closest approach, which is specific to the ion of interest, $z$ the charge of the ion, and $I$ the true ionic strength when all dissolved components are considered. The Haar–Gallagher–Kell[31] and Marshall and Franck[32] models were used to determine the thermodynamic properties and dissociation constant of $H_2O$ under our experimental conditions. Thermodynamic data for the aqueous species were obtained from refs. [33–36], and [37]. Extended parameters for NaCl in the Debye–Hückel equation were taken from ref. [38]. Thermodynamic data for the uranium oxides was taken from refs. [16] and [39]. The thermodynamic data for $MoO_2$, $MoO_3$, Ni, and NiO was taken from refs. [39,40] and [41].

Optimizations were conducted separately for $UCl_4{}^0$ data collected using $UO_2{}^{cryst}$ and $U_3O_8{}^{cryst}$. These optimizations also calculated the chloride activity in each experiment, which was used to construct Fig. 1. As the two aqueous uranium species are uncharged, the activity of uranium shown in Fig. 1 is equivalent to the molality of uranium. The Gibbs free energy changes for the reactions responsible for the formation of $UO_2Cl_2{}^0$ and $UCl_4{}^0$ were calculated using the standard Gibbs free energy of the other species involved in the reactions, namely $UO_2{}^{2+}$, $Y^{4+}$, and $Cl^-$. The Gibbs free energy change was then converted to a formation constant (logß) using the relationship $\Delta G^0 = -RT\ln K$. These formation constants are shown in Table 2. The uncertainty reported is that calculated using the OptimA program.

All calculated formation constants were fitted separately for $UO_2Cl_2°$ and $UCl_4°$ to the Bryzgalin–Ryzhenko equation of state[24]. Formation constants from ref. [21] for 100−250 °C were included in the optimization for $UO_2Cl_2{}^0$. The fit was then used to calculate equilibrium constants (logK) for the uranium oxide dissolution reactions. These are shown in Table 1. Uncertainties in the equilibrium constants were obtained by adjusting the measured uranium activity to the same pH and chloride activity using the stoichiometry of Reactions (1), (2), or (3), taking their standard deviation, and propagating this standard deviation to the equilibrium constant. The trend lines to the experimental data shown in Fig. 1 were calculated on the basis of these equilibrium constants with the exception of oxidizing data, the trend lines of which were calculated using equilibrium constants of 39.6 and 50.9 at 300 and 350 °C, respectively. These were fitted independently of the equilibrium constants reported in ref. [21].

**IOCG modeling**. The model depicting scenarios that could be encountered by a reduced uranium-bearing fluid in a magnetite-bearing IOCG deposit was made using the Hch software package[29]. The ore-forming temperature for magnetite-bearing IOCG deposits can exceed 500 °C[27,42,43]. However, a temperature of 400 °C was selected for our calculations so as to not significantly exceed the 250–350 °C temperature range of our experiments. The conclusions of this model would be the same for 500 °C except that the absolute concentration of dissolved uranium would be orders of magnitude higher. A pressure of 500 bars was used to maintain supercritical fluid conditions and to simulate conditions for an aqueous fluid at depth. Thermodynamic data sources for the dissolved aqueous species and solids used in the model are identical to those given in the preceding data optimization section of the supplementary information.

In the first model, which depicts a reduced fluid becoming progressively more oxidized, a fluid with $2m$ NaCl, $0.055m$ HCl, at 400 °C, 500 bar, and at a pH of 3.0 was saturated with respect to uraninite at a log $fO_2$ of $-35$ and had its log $fO_2$ progressively raised until a log $fO_2$ of $-15$ was achieved (Fig. 3a). Uraninite did not precipitate and the dominant aqueous uranium species in solution shifted from $UCl_4^0$ to $UO_2Cl_2^0$. Therefore, in the absence of an increase in pH, interactions with an oxidizing fluid, or wall rocks that may oxidize the fluid will not result in uranium deposition.

The initial fluid in the second model was more oxidizing with a log $fO_2$ of $-26$, equivalent to that of the magnetite–hematite redox buffer, but otherwise identical to that used in the first model. Progressive cooling of this fluid resulted in a change in speciation due to the more rapid decrease in stability of the $UCl_4^0$ species with decreasing temperature relative to the $UO_2Cl_2^0$ species, but no uraninite precipitation (Fig. 3b). The oxidizing and cooler conditions encountered at shallower depths in an IOCG system are therefore insufficient for uranium deposition in the absence of a pH increase.

The pH increase was considered in the third model. The same initial fluid as in the second model is titrated with NaOH to produce a pH increase. Uraninite deposition rapidly follows (Fig. 3c). Whereas this fluid is initially capable of dissolving considerable amounts of uranium, this metal cannot be transported at less acidic conditions.

The last model explores the impact of a temperature decrease on the preceding fluid, at reducing conditions (log $fO_2 = -35$). As the stability of $UCl_4^\circ$ is considerably more temperature dependent than that of $UO_2Cl_2^\circ$ (Fig. 2), uraninite deposition proceeds rapidly, despite the pH decrease accompanying NaCl dissociation (Fig. 3d). Therefore, these models demonstrate that in a magnetite-bearing IOCG system, temperature decreases at reducing conditions, or pH increases are necessary to remove the considerable amounts of uranium that could be present under reducing conditions.

**Data availability**. The data supporting the findings of this work are available within the article and its Supplementary Information files. All other relevant data supporting the findings of this study are available from the corresponding author on reasonable request.

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

## Acknowledgements

Funding for A.T.'s visit to the Los Alamos National Laboratory (LANL) was provided by a Seaborg Institute Summer Research Fellowship and is gratefully acknowledged. Additional support for A.T.'s stay at LANL was supplied by Natural Sciences and Engineering Research Council of Canada Alexander Graham Bell and Michael Smith Foreign Study Supplement Canada Graduate Scholarships. Research presented in this article was supported by the Laboratory Directed Research and Development program of Los Alamos National Laboratory under project number 20180007DR.

## Author contributions

A.A.M. conceived the research. A.T. developed the experimental method and conducted experiments with $U_3O_8$. A.A.M. performed experiments with $UO_2$. A.T. wrote the manuscript in collaboration with A.A.M. and A.E.W-J. Additional comments were provided by R.R., A.N, and H.X.

## Additional information

**Competing interests:** The authors declare no competing interests.

