## [Peer Review File · Nature Communications]

Reviewers' comments:

██
██

Reviewer #2 (Remarks to the Author):

The point of this study is nice and simple: The current consensus is that uranium can only be transported in oxidised form of U(VI) species in hydrothermal fluids; however this study reveals that U(IV) species, too, can transport uranium at elevated temperature in reduced fluids. The proposed responsible species is UCl₄.

The experimental and data analytical methods have been extensively used by the group over years for the investigation of metal solubility in hydrothermal fluids. The data are of high quality, and clearly show the solubility behaviour of the uranium oxide minerals. The proposed reduced species UCl₄ can be clearly seen from the slope of the linear trend of solubility plots – this is an unambiguous evidence of the predominance of this species. The subsequent discussion and modelling of uranium transport in the IOCG system, although a bit over-simplified, have demonstrated effectively the role of the newly-determined UCl₄ in the uranium transport in ore fluids at high temperature under reducing conditions, typical of IOCG magnetite stage.

The finding is simple but novel and elegant, and will have profound impact on our understanding of uranium transport and deposition in hydrothermal fluids relevant to ore formation, nuclear power plant management and waste treatment. The manuscript is well-written and easy to understand with few errors. I strongly recommend publication of this exciting and important study in Nature Communications.

Some minor comments:

1. Line 35: change 'oxidizing' to 'oxidized'.
2. Line 36-37: this sentence better be: "Our experiments have identified new uranium chloride species (UCl₄) that is stable under reducing conditions"
3. In the supplementary document, compile a table or make a plot to show the logfO₂ values of the MoO₂/MoO₃ and Ni/NiO buffer for all experimental temperatures (250-350C)
4. Line 192-193; these two sentences are unnecessary and should be deleted: " We expected to obtain very similar formation constants for UCl₄ from reaction(2). This was the case."
5. Line 337: Need to specify that reaction equilibrium has been reached for the duration of experimental time of 7 days?
6. Line 365-366: I think that the molality of HCl is better determined by initial concentration of HCl added, not the pH measured AFTER each experiment, though the difference may be insignificant.

Reviewer #3 (Remarks to the Author):

Very interesting manuscript that challenges the long-standing assumption that U⁴⁺ is immobile under reducing conditions. The results presented in the paper, coupled with published spectroscopic study (Migdisov et al., 2018), provide convincing evidence for the presence of uranium chloride species in reduced solutions at elevated temperatures. As noted by the authors, the presence and concentration of these uranium species will force re-examination of several uranium ore deposit models (unconformity-type, uranium-bearing IOCG, metasomatic uranium, to name a few) but also have broader implications for nuclear fuel storage, acid mine/tailings drainage, etc. For example, the results and modelling in the paper for magnetite-group IOCG deposits provides a more plausible explanation for uranium sourcing than the currently-accepted model that invokes input of uranium-bearing, oxidized fluids as the uranium source (especially when the relationships we have noted indicate that oxidation seems to occur after precipitation of

uranium and other typically "immobile" elements).

Recommend publication with minor revisions (see annotated MS Word document) as the research is novel, should have a large impact in geosciences and will be of interest to a broad audience.

Regards, Eric Potter

Responses to Reviewers

Reviewer #1:

[REDACTED]

[REDACTED]

[REDACTED]

[REDACTED]

[REDACTED]

[REDACTED]

[REDACTED]

[REDACTED]

[REDACTED]

[REDACTED]

[REDACTED]

[REDACTED]

[REDACTED]

[REDACTED]

[REDACTED]

[REDACTED]

[REDACTED]

[REDACTED]

[REDACTED]

[REDACTED]

[REDACTED]

[REDACTED]

[REDACTED]

[REDACTED]

[REDACTED]

[REDACTED]

[REDACTED]

[REDACTED]

[REDACTED]

[REDACTED]

[REDACTED]

[REDACTED]

[REDACTED]

[REDACTED]

[REDACTED]

[REDACTED]

[REDACTED]

[REDACTED]

[REDACTED]

[REDACTED]

[REDACTED]

A large rectangular area of the page is completely redacted with a solid black fill, obscuring all text and graphics that would otherwise be present.

Reviewer #2:

1. *Line 35: change 'oxidizing' to 'oxidized'.*

The word "oxidizing" has been changed to "oxidized".

2. *Line 36-37: this sentence better be: "Our experiments have identified new uranium chloride species (UCl₄) that is stable under reducing conditions"*

The chloride species has been added in brackets and the sentence now reads "Our experiments have identified a new uranium chloride species (UCl₄) that is more stable under reducing than oxidized conditions."

3. *In the supplementary document, compile a table or make a plot to show the logfO₂ values of the MoO₂/MoO₃ and Ni/NiO buffer for all experimental temperatures (250-350C)*

An additional table has been added to the supplementary document that provides all logfO₂ values for the MoO₂/MoO₃, Ni/NiO, and Co/CoO buffers from 250 to 350 °C.

4. *Line 192-193; these two sentences are unnecessary and should be deleted: " We expected to obtain very similar formation constants for UCl₄ from reaction(2). This was the case."*

The two sentences have been removed.

5. *Line 337: Need to specify that reaction equilibrium has been reached for the duration of experimental time of 7 days?*

The sentence has been modified to read "The duration of each experiment was 7 days, which allowed the experiments to reach equilibrium."

6. *Line 365-366: I think that the molality of HCl is better determined by initial concentration of HCl added, not the pH measured AFTER each experiment, though the difference may be insignificant.*

As the pH of the experiment can change during the experiment, albeit typically not by a great amount, and the pH of the solution is controlled by the concentration of HCl, we report the molality of HCl following the experiments, which better reflects the pH and concentration of HCl in the experimental solution at equilibrium.

Reviewer #3:

1. *suggest to change to something reflecting that low temperature data were extrapolated to higher temperatures and no Cl-rich solutions tested*

The abstract of the paper serves as a succinct summary of the paper and, as later in the text we explicitly discuss the issue of the absence of thermodynamic data at high temperature, particularly for Cl-rich solutions, we have left the phrase in its shortened form so as to make the abstract easier to understand for a general audience.

2. *Suggest a bit stronger wording for the last sentence as these are very important results.*

We have modified the wording at the end of the abstract so as to highlight the impact of our results. However, we cannot propose with any confidence that a redesign of long term nuclear waste facilities is necessary. This is because, if groundwater were to infiltrate a storage facility, it is not expected that the temperature of this infiltration would exceed 150 °C, which is below the temperature for which our data suggest that reduced uranium chloride species can occur in significant concentrations. Accordingly, we have not considered this potential application in our study.

3. *Should this not be low temp fluids?*

We use the term high temperature, as low temperature geochemistry is considered to pertain to temperatures of less than 100 °C. To make this clearer, we have adopted the suggestion from Comment 4 and provided the "high temperature" range in brackets.

4. *should quantify "high temperature" (100 to >400 oC)*

We have added the temperature range considered in brackets.

5. *would make more sense as (a) as you discuss the UO2 experiments first*

We have switched the order of (a) and (b) in the figure.

6. *give some exact values (e.g. x ppm at 300oC with Cl activity = y).*

We have added an example in brackets as follows "(e.g. ~4ppm at 400 °C with aCl⁻ = 0.10 and a pH of 2.5)".

7. *lower T limit for IOCG systems: 800 – 250 oC as system evolves from Na to Ca-Fe (magnetite) to K-Fe (magnetite) then to K-Fe (hematite+/-sericite+/-carbonate) alteration.*

K-Fe (magnetite) alteration is 450-350 oC, but these fluids scavenged U and other metals from the country rocks during Na (600-300oC) and Ca-Fe (800-450 oC) metasomatism/alteration.

We agree that uranium mobilization could occur at temperatures greater than 400 °C but chose to work with a lower temperature so as to avoid having to extrapolate our solubility data too much beyond the temperature range of this study (250-350 °C). The reason for this is that the trends in our data suggest that uranium solubility may increase exponentially with temperature, but this is a hypothesis that remains to be tested.

8. *Would also explain why the uraninite from these systems contain moderate to high Th contents that mirror magmatic-derived compositions, ranging from 0.25 to 12.9 wt. % Th. (Potter, E.G., Corriveau, L., Montreuil, J-F., Yang, Z., and Comeau, J-S, 2014. Geochemical signatures of uraninite from iron oxide-copper-gold (IOCG) systems of the Great Bear magmatic zone, Canada; Geological Survey of Canada, Open File 7545, 1 poster. doi:10.4095/293702)*

We agree with this comment.

9. *needs slight re-wording. Suggest something like: One model for the genesis of unconformity-type U deposits invokes leaching of U from reduced basement rocks containing elevated concentrations of uraninite (UO₂) (ref.23). As shown by our experimental work, leaching of uranium by the strongly acidic pH, Cl-rich brines documented in the vicinity of these deposits at temperatures 130°–220°C (Richard et al., 2010, 2012) would favor efficient U extraction and transport.*

Although we would like to employ stronger wording, the accepted model for the formation of unconformity-type uranium deposits, which involves oxidizing fluids bringing uranium to reduced sites of deposition, is so ingrained in the literature that no one has attempted to develop a model in which uranium is sourced from the basement rocks. Studies such as ref. 23 have identified potential sources of uranium in the basement rocks, but only hint at their leaching, and studies such as ref. 25, while invoking leaching of these sources have stopped short of developing a comprehensive model for unconformity-type deposits involving a basement source and reduced ore fluids. All we can do, given the nature of our study, is point out the need to re-evaluate the existing model in light of the results of our experiments.

10. *true but also cite work of Romberger (1984) who did work at temperatures up to 300 oC Romberger, S.B. (1984): Transport and deposition of uranium in hydrothermal systems at temperatures up to 300oC: geological implications, In De Vivo B. (ed.), Uranium geochemistry, mineralogy, geology , exploration and resources; The Institution of Mining and Metallurgy, p. 12-18.*

The study of Romberger (1984) does not present any original experimental findings and uses existing low temperature thermodynamic data to evaluate uranium solubility and speciation. Moreover, this author does not provide the sources of his thermodynamic data nor methods of extrapolation. As the accuracy of such extrapolations is commonly quite poor, we cannot refer to the predictions of Romberger with sufficient confidence to include them in this paper.

11. *fits nicely with the temperature of the fluids as the K-Fe (magnetite) alteration assemblage characteristic of magnetite-group IOCG systems form (i.e., 450-350oC)*

This is why we chose to evaluate the mobility of uranium in an IOCG setting at 400 °C, but higher or even slightly lower temperatures could also warrant modelling, as noted here and in our response to Comment 7.

12. *and which might explain why we don't see any U in the higher temperature Na and Ca-Fe alteration assemblages.*

We agree with this comment.

13. *does just fall into the lower pressure range of IOCG districts – see Barton (2014): Iron Oxide(-Cu-Au-REE-P-Ag-U-Co) Systems; Treatise on Geochemistry 2nd Edition*
<http://dx.doi.org/10.1016/B978-0-08-095975-7.01123-2>

We did our modelling at a lower pressure so as to minimize extrapolations from our experiments, which were conducted at saturated water vapor pressure, but this pressure does fall in the range expected of IOCG deposits, as is noted in Barton (2014).